# A Survey on Vaccination and Disease Occurrence in Municipal and Non-Profit Animal Shelters in Portugal

**DOI:** 10.3390/ani13172723

**Published:** 2023-08-27

**Authors:** Sara Marques, Eduarda Gomes-Neves, Cláudia S. Baptista, Francisca R. Pereira, Adélia Alves-Pereira, Pedro Osório, Alexandra Müller

**Affiliations:** 1Instituto de Ciências Biomédicas Abel Salazar (ICBAS), University of Porto, Rua Jorge Viterbo Ferreira 228, 4050-313 Porto, Portugal; emneves@icbas.up.pt (E.G.-N.); csbaptista@icbas.up.pt (C.S.B.); up201810076@edu.icbas.up.pt (F.R.P.); 2Centro de Investigação em Biodiversidade e Recursos Genéticos (CIBIO), Laboratório Associado em Biodiversidade e Biologia Evolutiva (InBIO), University of Porto, Campus Agrário de Vairão, Rua Padre Armando Quintas No. 7, 4485-661 Vila do Conde, Portugal; 3Centro de Estudos de Ciência Animal (CECA), Instituto de Ciências, Tecnologias e Agroambiente (ICETA), Laboratório Associado AL4AnimalS, University of Porto, 4050-313 Porto, Portugal; 4Shelter Veterinarian, 4050-313 Porto, Portugal; adeliamedicina.abrigo@gmail.com (A.A.-P.); pedrocunhaosorio@gmail.com (P.O.)

**Keywords:** questionnaire, rescue shelter, veterinary medicine, Portugal, canine, feline, companion animals, length of stay, infections

## Abstract

**Simple Summary:**

A common belief is that animal shelters still have a high occurrence of disease. Few studies are available describing the current situation in animal shelters. We characterized this issue with an online questionnaire sent to animal shelters in Portugal. Apart from municipal animal shelters, other types of animal shelters were also included, such as associations which are non-profit organizations. The response to this questionnaire was voluntary. More veterinarians responded for municipal shelters, whereas more non-vets answered the questionnaire on behalf of associations. Preventive measures such as worming, vaccination and the use individual animal medical records were widespread among both shelter types. Many shelters indicated keeping animals for over 1 year. This excessive length of stay should be reduced, for example by better training of staff and the availability of adequate shelter management software. Puppy re-vaccination every 2 to 4 weeks was indicated by both shelter types and final vaccination at 16 weeks was predominantly indicated by associations. Most adult animals were revaccinated annually. The three most reported diseases were parvovirus and mange in dogs, cat flu and panleukopenia in cats, and ringworm in both species. Additional information on disease occurrence should be obtained by objective monitoring.

**Abstract:**

Few studies are available describing animal shelters in Portugal. The aim was to characterize prophylactic measures and disease occurrence in shelters with a questionnaire. The response rates of 67 shelters (42 municipal shelters, 25 associations) were compared by the Fisher’s exact test. More veterinarians answered for municipal shelters (98%) than for associations (40%; *p* < 0.001). Over 80% of the respondents indicated using individual medical records and routine prophylaxis. Excessive length of stay for dogs was reported by 54% of associations and 33% of municipal shelters. Management tools should be promoted to improve the situation. Puppy vaccinations were similar and a final vaccination at 16 weeks was indicated by >33% of shelters. Annual revaccination of dogs was reported more frequently by associations (88%) than municipal shelters (55%; *p* = 0.02). The three most reported diseases were parvovirus and mange in dogs, upper respiratory disease and panleukopenia in cats, and dermatophytosis in both species. Similar response rates for diagnostic options were obtained by both shelter types, except for distemper. Testing for feline retroviruses was indicated by most shelters (>69%), but only a few (<24%) confirmed positive test results. Clinical diagnoses should be complemented by testing. Additional information on disease occurrence should be obtained by objective monitoring.

## 1. Introduction

Animal shelter medicine is a relatively recent veterinary area that has contributed substantially to optimizing the various aspects of the management of shelters [1,2,3,4]. Portuguese legislation regarding governmental animal shelters was changed in 2016, coming into force in 2018, advocating the modernization of the formerly denominated municipal kennels and now called official animal collection centres [5,6]. These centres, herein denominated municipal shelters, continue to be managed by city councils and are responsible for safeguarding public health and stray animal population control. This new law reinforced the use of methods other than euthanasia for population control. This prohibition of euthanasia and the limited capacity of existing municipal shelters increased the risk of overcrowding and related welfare issues. Therefore, the law invokes the collaboration between municipal and non-governmental animal shelters to control stray animals, to educate for responsible ownership and to promote adoption [5,6]. Non-governmental shelters consist predominantly of non-profit organizations, herein denominated animal associations. These also do operate in their own dedicated buildings and may additionally have fostering animals in private homes or other systems in place. Both types of shelters have to be registered by the National Authority for Animal Welfare and to comply with functional and structural requirements, which include medical and prophylactic programmes that are not further specified [7]. Otherwise, the funding, staffing, objectives and organization of these two main shelter types frequently differ, leading to potential differences in infectious disease occurrence [8,9].

Managing sheltered animal populations is complex as most shelters consist of open populations, several species, unpredictable intakes and releases resulting in variable lengths of stay. All these factors make the management of infectious diseases in shelters challenging [1,9,10]. Ideally, each shelter should have its own healthcare programme, which should include a physical examination and vaccination on admission, internal and external parasite control and disease testing among others [1]. In Portugal, mandatory sanitary measures have been established for municipal shelters and apply only to animals leaving the municipal shelter, e.g., via adoption. These measures consist of identification by microchip, adequate parasite control and mandatory vaccination, which in Portugal only relates to the rabies vaccination of dogs [6,11]. Specific shelter standards or health care programmes are non-existent in Portugal and are currently being developed. Recently, universities started to offer continuing professional development on shelter medicine. Only a few shelter management softwares are commercially available, and these do not allow for the extraction of adequate information, e.g., on length of stay.

Vaccination is an effective prophylactic tool to minimize the occurrence of infectious diseases. Vaccine-preventable diseases include parvovirus, distemper, infectious hepatitis in dogs and panleukopenia in cats [12,13,14]. Infections such as leptospirosis, canine infectious tracheobronchitis and feline upper respiratory diseases caused by herpes- and caliciviruses, cannot be completely prevented; however, vaccination contributes to disease control by reducing clinical manifestation, shedding and transmission [12,15]. Current vaccination guidelines strongly advocate for a high population immunity by vaccinating as many animals as possible and vaccinating at intake to the shelter [12,13,14]. In the past, municipal shelters vaccinated only against rabies, but today, different vaccines and vaccination schedules are being practised in all shelter types. The availability of funding may impact the availability of sufficient vaccines in economically vulnerable shelters, e.g., some may have to prioritize between feeding and prophylactic measures. Details on the use of vaccination against other than rabies are currently unknown in shelters in Portugal.

Regarding animal shelters, the common conception has been mentioned that overcrowding, underfunding and poor facilities are leading to a high occurrence of disease [1]. The occurrence of infectious disease in shelters has been investigated from many angles, mostly in the US [9,16]. Some investigations focused on known infections [17,18,19], others on emergent agents or outbreaks [20,21], or on clinical syndromes such as respiratory [22,23,24] or gastro-enteric infections [25]. Whether the situation is similar in Portugal is unknown, as the occurrence of infectious disease in shelters is largely unreported. This may be due to multiple reasons such as the performance of different diagnostic assays [26] as well as a lack of monitoring and reporting systems [1]. If we examine the personal experience of shelter veterinarians in Portugal, it is clear that some shelter conditions have improved in the past decades while others require urgent interventions; however, scientific data are lacking.

The aim of this work was to characterize common prophylactic measures and the occurrence of infectious disease in animal shelters at national level, and to compare municipal shelters with animal associations. 

## 2. Materials and Methods

### 2.1. Questionnaire

To characterize animal shelters in Portugal, a questionnaire was developed, containing 11 sections on several aspects. It was reviewed by five external shelter veterinarians and their minor comments were incorporated in the final version. The questionnaire was approved by the Ethics Committee of ICBAS-CHUP and delivered on an online platform between 1 February and 31 March 2021 with an email reminder being sent on 15 March. According to the instructions, the questionnaire should be filled in by the responsible veterinarian. Responses were completed during the COVID-19 pandemic. Partial results have been described previously [8,27,28,29]. The cover letter and the sections of the questionnaire presented here are available as Appendix A.

General aspects included the description of the type of shelter (non-profit shelter with adoption, sanctuary, municipal shelter, inter-municipal shelter, other). Other questions focused on whether the respondent was a veterinarian (yes, no) and on which animals were admitted to the shelter (dogs, cats), the existence of individual medical records (yes, no), internal parasite control (yes, no), external parasite control (yes, no), vaccination (yes, no) and financial restrictions for such routine prophylactic measures (yes, no). The type of veterinary assistance excluding neutering allowed more than one option (staff veterinarian, services of the veterinary clinic, consulting or visiting veterinarian, and other). The proportion of animals that were in the shelter for more than one year was assessed separately for dogs and cats (<5%, 5–10%, >10–25%, >25–50%, >50%, non-applicable). We selected the results only for the category “>50%” of the animals residing in shelter for more than one year as being a proxy for an excessive length of stay.

Specific aspects related to infectious diseases were evaluated by questions on vaccination, disease occurrence and diagnosis as follows. Questions on vaccination included available vaccines for dogs (rabies, DHPPi + L, kennel cough, vaccines unavailable) and cats (RCP, FeLV, rabies, vaccines unavailable). Vaccination schedules for puppies and kittens were assessed for both species together (repeated every 2–4 weeks, final vaccination at 16 weeks or older) and adult revaccinations were considered separately for dogs and cats (annual, every 3 years, no revaccination). The occurrence of specific infectious diseases within the previous year was assessed for each species. For dogs, it included canine distemper, parvovirus, canine coronavirus, canine infectious tracheobronchitis (also known as kennel cough), mange and dermatophytosis. In addition for cats, the following were assessed: feline panleukopenia (also known as feline parvovirus, FPV), feline upper respiratory diseases (FURD) including feline herpes- and calicivirus (also known collectively as cat flu), feline infectious peritonitis (FIP), mange and dermatophytosis. The English term “mange” was used for the Portuguese term “sarna” in the questionnaire. No distinction between dog-specific *Demodex* or zoonotic *Sarcoptes* was made, and the term included both. Response options to these infections included absence of disease, 1or 2 cases, >2 cases, (at least 1) outbreak, non-applicable and unknown. Cases were defined as affecting individual animals at different time points, as opposed to outbreaks, where multiple animals are affected within the same time period. Due to the low number of responses, the options “1 or 2 cases” and “>2 cases” were merged into the category “sporadic cases”. The options “unknown” and “non-applicable” were also merged and as they were considered non-informative, they represent the remainder of 100% of the responses in the graphs, i.e., they were not explicitly shown. The main diagnostic method for specific infectious diseases (distemper, parvovirus and coronavirus infections in dogs; panleukopenia, cat flu and FIP in cats) was assessed by the response options: “clinical diagnosis”, “rapid test/laboratory” and “necropsy”. Clinical diagnosis included physical examination and history without diagnostic laboratory tests. Questions with multiple options also included the option “non-applicable”. The questionnaire and cover letter is available as Appendix A.

### 2.2. Data Analysis

The percentages of the responses of animal associations and of municipal shelters were displayed graphically. Not all shelters hosted both species, dogs and cats. Therefore, for species-specific questions, the percentage of respondents was calculated by selecting shelters hosting dogs (24 associations and 42 municipal shelters) or cats (15 associations and 26 municipal shelters). The Fisher’s exact test was used to investigate if there was an association between response options and shelter type (association, municipal shelter). The availability of vaccines was compared individually, for each vaccine type, between associations and municipal shelters. Significance was considered at *p* < 0.05. The Fisher’s exact test was run in the online calculator MedCalc [30]. Numbers of cats and dogs were compared between shelter types using the independent-samples Mann–Whitney U test in the SPSS version 26.0 [31].

## 3. Results

### 3.1. General Aspects

A total of 67 animal shelters completed the questionnaire: 42 municipal shelters and 25 associations. Regarding the shelter type, municipal and inter-municipal shelters were merged to represent “municipal shelters”, as both are governmental. Inter-municipal shelters represent two or more small municipalities. The remainder, i.e., non-profit shelters with adoption and others, were merged to represent “associations”. The three respondents of sanctuaries were excluded from the analysis as they did not complete the questionnaire. The response rate was 43% (42/97) from municipal shelters and 39% (25/65) from associations (d.f. 1, *p* = 0.5). The questionnaires were completed by 51 veterinarians and 16 non-veterinarians. The proportion of questionnaires filled in by veterinarians was significantly higher for municipal shelters (98%, 41/42) than for associations (40%, 10/25; *p* < 0.001). Similar response rates were obtained by both shelter types regarding the species admitted. Dogs were admitted by 96% (24/25) and 100% (42/42), and cats by 60% (15/25) and 62% (26/42) of associations and municipal shelters, respectively. The median number of animals was significantly higher in Associations than in Municipal shelters [7]. The median number of dogs in associations was 225 compared to 60 in municipal shelters (U = 271.5; *p* = 0.04). The median number of cats was 48 in associations and 11 in municipal shelters (U = 86; *p* = 0.03).

Over 80% of the respondent shelters indicated the use of individual animal medical records (20/25 associations, 38/42 municipal shelters, *p* = 0.3), and routine prophylactic measures such as internal (24/25 associations, 42/42 municipal shelters, *p* = 0.4), and external parasite treatment (24/25 associations, 39/42 municipal shelters, *p* = 1), and vaccination (24/25 associations, 35/42 municipal shelters; *p* = 0.2; Figure 1). Financial restrictions for routine prophylaxis were indicated by 36% (9/25) of associations and 24% (10/42) of municipal shelters (*p* = 0.3).

Veterinary care differed between shelters (Figure 2). Over 90% (38/42) of the municipal shelters indicated relying on staff veterinarian(s), compared to 44% (11/25) of associations (*p* < 0.001). The reliance on visiting veterinarians was indicated by 36% (9/25) of the associations and none of the municipal shelters (0/42; *p* < 0.001). The services of veterinary clinics were indicated by 76% (19/25) of associations and 41% (17/42) of municipal shelters (*p* = 0.09).

Regarding the excessive length of stay for dogs, 54% (13/24) of associations and 33% (14/42) of municipal shelters indicated that more that 50% of their dogs were in the shelter for more than 1 year (*p* = 0.1). Regarding cats, more associations (60%, 9/15) than municipal shelters (12%, 3/26; *p* = 0.003) reported an excessive length of stay. Results for the other response categories of animals remaining in shelters more than 1 year are represented in Table 1.

### 3.2. Vaccination 

#### 3.2.1. Vaccine Availability 

The availability of dog and cat vaccines was similar between associations and municipal shelters (Figure 3a,b; *p* > 0.05), except for the rabies vaccine for cats, which was less available in associations (33%, 5/15) compared to municipal shelters (69%, 18/26; *p* = 0.049). Core vaccines (DHPPi + L for dogs and RCP for cats) were available at more than 69% of shelters and optional vaccines (KC for dogs and FeLV for cats) at less than 33%.

#### 3.2.2. Vaccination Schedules

Vaccination schedules were assessed for young animals (puppies and kittens) and for adults (Figure 4). Puppy and kitten vaccination schedules were similar among both shelter types (Figure 4a). Repeated vaccinations every 2 to 4 weeks were reported by >60% of both shelter types (15/25 associations, 27/42 municipal shelters), and a final vaccination at 16 weeks or older was indicated by 52% (13/25) of associations and 33% (14/42) of municipal shelters (*p* = 0.2). Revaccination of adult animals every 3 years was reported by less than 20% of both shelter types (Figure 4b,c). Annual revaccination of dogs was reported more frequently by associations (88%, 21/24) than municipal shelters (55%, 23/42; *p* = 0.01). In cats, annual revaccination was also more frequently reported by associations (73%, 11/15) compared to municipal shelters (46%, 12/26), however this was not statistically significant (*p* = 0.1). Regarding the revaccination of cats, the option “non-applicable” was selected more frequently (35%, 9/26) by municipal shelters than associations (0/15; *p* = 0.02).

### 3.3. Occurrence of Infectious Disease

The responses to the question on the occurrence of specific infectious diseases in dogs and cats within the past 12 months are shown in Figure 5. Regarding infectious diseases in dogs, the occurrence of outbreaks as well as sporadic cases was reported similarly between both shelter types (Figure 5a,b). An exception was dermatophytosis, where more sporadic cases were reported by municipal shelters (45%, 19/42) compared to associations (17%, 4/24; *p* = 0.03). The answer “absence of disease” was similar between shelter types except for distemper, being more frequently chosen by associations (96%, 23/24) compared to municipal shelters (74%, 31/42; *p* = 0.04). Regardless of shelter type, among all 66 respondents, the most frequently reported infection in dogs was canine parvovirus (62%, 41/66), followed by mange (50%, 33/66), dermatophytosis (37%, 24/66), kennel cough (27%, 18/66), canine coronavirus (12%, 8/66) and distemper (9%, 6/66). Despite the occurrence of canine distemper being reported by six municipal shelters and none of the associations, this difference was not statistically significant (*p* = 0.08). The reported occurrence of infectious disease in cats was similar between shelter types (Figure 5c,d). The answer “absence of disease” was similar for all diseases except for FIP, where absence was indicated by 80% (12/15) of associations compared to 39% (10/26) of municipal shelters (*p* = 0.02). Regardless of shelter type, the most frequently reported infection by the 41 shelters housing cats was cat flu (68%, 28/41), followed by panleukopenia (44%, 18/41), dermatophytosis (22%, 9/41), FIP (15%, 6/41) and mange (10%, 4/41). 

### 3.4. Main Diagnostic Methods

The question on the main diagnostic methods for common infectious diseases had three response options: “clinical”, “rapid test” and “necropsy”. The latter option was not selected by any respondent. Similar response rates for each diagnostic option in dogs and cats were obtained by both shelter types, except for distemper, where more municipal shelters (48%, 20/42) reported the use of clinical diagnosis, compared to 17% (4/24) of associations (*p* = 0.02, Figure 6).

The routine use of rapid tests to diagnose feline retroviruses was indicated by 87% (13/15) of associations and 69% of municipal shelters (18/26; *p* = 0.3, Figure 7). Of the shelters using rapid tests, 23% of associations (3/13) and 6% of municipal shelters (1/18) reported confirming positive test results by using another rapid or laboratory test. The differences between associations and municipal shelters were not statistically significant (*p* = 0.3). 

## 4. Discussion

The collaboration between municipal shelters and associations is a requirement in Portugal. No further information on management or the occurrence of diseases is available. A questionnaire was developed to characterise different aspects of animal shelters [8,27,28,29], including vaccination and infectious diseases. Every shelter healthcare program should include a strategy for disease response and prevention. Incorrect or non-existent vaccination schedules in animal shelters can impair the control of infectious disease and even promote outbreaks [1,4,32].

Similar to others, our survey demonstrated that prophylactic measures were common [9]. Over 80% of the respondent animal shelters indicated the use of individual animal medical records and the application of routine prophylactic measures such as vaccinations and external and internal parasite control. Importantly, approximately one-third of both associations and municipal shelters indicated financial restrictions for routine prophylaxis, requiring better funding for healthcare programs. The type of veterinary assistance also differed, staff veterinarians being more common in municipal shelters and consulting (visiting) veterinarians in associations. This may explain some of the different response rates between shelter types on vaccination, disease occurrence and diagnosis. 

Regarding vaccination, current recommendations are to vaccinate all adult animals at or before shelter intake [1,4]. We did not evaluate the timing of vaccination, but similar to others found a high vaccination coverage [9]. Respondents indicated that over two-thirds of the sheltered animals had been vaccinated, regardless of shelter type. The use of core vaccines (DHPPi + L for dogs and RCP for cats) was indicated by over two-thirds of the respondent shelters, whereas optional vaccines (KC for dogs and FeLV for cats) were only reported by one-third, with similar response rates for associations and municipal shelters. One interesting difference was related to the rabies vaccine for cats, which was more available in municipal shelters compared to associations. This was unexpected because according to the national legislation rabies vaccination is mandatory for dogs but not cats [11,33]. As Portugal has been free from terrestrial rabies since 1961, there are no local mandates for rabies vaccination, so this vaccination is unnecessary, and financial resources could be used otherwise [34].

Today, vaccination protocols are designed for each animal individually taking into consideration their lifestyle and other factors [1,12,13,14]. The probability of exposure and the potential consequences of infection require clearly defined shelter vaccination programs. The recommendations for puppies and kittens entering a shelter indicate that core vaccination may be started as early as 4–6 weeks of age, and revaccination should be every 2 weeks until at least 16 weeks of age, if still in the shelter [1,12]. Regarding primo vaccinations in puppies and kittens, our questionnaire did not cover the age at first vaccinations because we assumed that these and litters would be admitted at any age, and that age would have to be estimated due to unknown history. Regarding the frequency of vaccinations, the response rate between associations and municipal shelters was similar. Over two-thirds indicated repeatedly vaccinating every 2 to 4 weeks. Although more associations than municipal shelters reported applying a final vaccination at 16 weeks or older, this was not statistically different. We hypothesize that animals in municipal shelters are re-homed earlier or transferred at an early age to associations that then promote adoptions, according to the current legal framework [5,6]. Thus, associations could be hosting more animals that are 16 weeks or older. 

Current recommendations advocate revaccination with live attenuated core vaccines every 3 years instead of annual boosters [12]. However, in companion animals, annual revaccinations are still common practice, especially as there are still some commercial vaccines indicating annual vaccination on their label. Annual revaccination was much more common than triannual and reported more frequently by associations than municipal shelters. This may be explained by the length of stay of animals in shelters, where especially associations seem to struggle. Our results indicate that approximately every second association keeps more than 50% of their dogs and cats for over 1 year. The situation is slightly better regarding municipal shelters, but is still far from ideal. The excessive length of stay could be a key factor underlying high stocking rates in animal shelters in Portugal. Here, managing the intake rate is difficult and euthanasia for space rather than illness, injury or danger to others from behaviour is not an option. Therefore, shelter resources are consumed by taking care of the existing animals. Adoptions and outreach to promote live outcomes should be reinforced. The situation in Portugal likely reflects a lack of knowledge and/or difficulties in implementing established shelter management strategies, e.g., such as the ASV Guidelines for Standards of Care in Animal Shelters [4]. These as well as other management tools, such as adequate shelter software and professional training on concepts such as capacity for care and strategies to reduce length of stay should be reinforced. We expect that the recently developed training courses will contribute to improving the current situation. Bearing in mind a generally high vaccination coverage, even in young animals, and a reduced turnover due to the long in-shelter stay of a considerable proportion of animals, we expected to find a relatively low occurrence of vaccine-preventable diseases. 

The occurrence of infectious diseases was assessed semi-quantitatively. The respondents were asked to indicate if specific diseases were absent, sporadic or if outbreaks had occurred in the previous 12 months. Disease occurrence was similar between both shelter types, with three exceptions. Canine dermatophytosis, canine distemper and FIP in cats were more frequently reported by municipal shelters than by associations. This was rather unexpected because the prevalence of these infections should not be influenced by shelter type [35,36,37,38,39]. Unspecific clinical signs are common in these diseases; therefore, diagnosis should be supported with adequate testing [1]. Financial restrictions could be an important factor limiting the access to laboratory testing of associations, as they are more dependent on donations whereas most municipal shelters can rely on regular public funding [8]. Another explanation for the differences in disease occurrence between shelter types could be the professional training of respondents as well as the type of veterinary assistance. Nearly all the respondents from municipal shelters were veterinarians, most likely staff veterinarians, compared to less than half of the associations, which may have selected respondents with other roles such as management or fund-raising. These could have been less capable of answering the technical disease-related questions in the survey, introducing some bias. 

Regardless of shelter type, the two most frequently reported infections in dogs were canine parvovirus followed by mange. Despite being vaccine-preventable, parvovirus is still common in shelters, which can be explained by the environmental resistance of the virus as well as the interference of maternal immunity during the primo-vaccination [12,40]. This strengthens the need to vaccinate all dogs on intake and to vaccinate puppies repeatedly, ideally until the age of 16 weeks or older. Regarding mange, unfortunately we were unable to evaluate the occurrence of zoonotic *Sarcoptes*, also known as scabies. This should be included in further studies. In cats, the two most frequently reported diseases were feline upper respiratory disease (cat flu) and panleukopenia. Interestingly, these are vaccine-preventable, as the common trivalent core vaccines of cats include feline herpesvirus, calicivirus and feline panleukopenia. Vaccine immunity against herpes and calicivirus is not sterile; cats can still harbour the virus, but clinical manifestations should be less severe in vaccinated cats. In shelter environments, there are many stressful conditions for cats, such as being confined with other unknown cats, hearing dogs barking, etc., which may impair the immune response, leading to the manifestation of clinical disease [1,10,41]. Feline panleukopenia is caused by a parvovirus, which, similarly to dogs, is difficult to eradicate, especially in the shelter environment. Dermatophytosis is the third most frequently reported infection in both dogs and cats. While of limited health consequence, it is a zoonotic disease which is common in shelters and can be difficult to control. Treating dermatophytosis to clear the infection is extremely lengthy, adding to cost and time in shelter, particularly of potentially more adoptable kittens and puppies. Cats are carriers and disease manifestation is exacerbated by stress and poor grooming [35]. Therefore, a systematic screening of cats has been suggested for shelters [1]. 

Any estimate of disease occurrence is influenced by the sensitivity and specificity of the diagnostic approaches used. We assessed the diagnosis by enquiring about the main diagnostic methods, contrasting clinical diagnosis with the use of diagnostic tests. Affordable rapid patient side assays, such as rapid immunomigration tests, are available for distemper, parvovirus, canine coronavirus and feline retroviruses. These diseases should be diagnosed also by testing and not only clinically. Both shelter types indicated similar response rates for each diagnostic approach, except for distemper, where municipal shelters reported a higher use of clinical diagnosis. This, as mentioned above, could be related to the professional background of the respondent. Any diagnosis of distemper based solely on clinical signs may have led to an overestimation of CDV by municipal shelters, or alternatively, to an underestimation by associations. 

The diagnosis of feline retroviruses relies on the use of in-clinic tests, such as rapid immunomigration. In general, screening of cats for FIV and FeLV is recommended based on shelter resources and local prevalence [10,26]. If routine testing is not possible, at least high-risk cats, including those entering group housing and sick cats, should be tested [4,10,26]. The ASV recommends confirmatory testing of healthy shelter cats according to the AAFP guidelines, which are similar to owned cats [4,10,26]. FIV-seropositive kittens should be retested to exclude interference of maternal antibodies. Retesting is also considered of particular importance for FeLV, to identify false positive results and also to distinguish between regressive and progressive infections [42,43]. The routine use of rapid tests to diagnose feline retroviruses was indicated by most shelters; however, only few reported confirming positive test results. Our questionnaire did not evaluate under which circumstances testing and re-testing were performed. In general, the seroprevalence of FIV in Portugal is highly variable ranging from 3% in owned cats to 22% in feline shelters, whereas FeLV prevalence ranged between 6 and 12% [44,45,46,47]. In our opinion, taking into account the variable prevalence of retrovirus infection, all positive tests results should be confirmed at some point.

Although this questionnaire enabled us to obtain information on prophylactic measures and disease occurrence in animal shelters, the study had some limitations. Selection bias likely occurred, in that only better-managed shelters decided to respond. Many municipal shelter veterinarians claim knowing associations that do not vaccinate animals at all, and where the sanitary and welfare conditions are questionable. Therefore, our results are unlikely to be representative of all animal shelters. In addition, recall bias could have occurred as some questions required information from the previous 12 months. The questionnaire structure presented too many questions that made analyses difficult. Furthermore, the use of vague terminology reduced the usefulness of some answers, such as the term “mange”, making it impossible to assess dog-specific *Demodex* or zoonotic *Sarcoptes* separately. Further questionnaires should be more direct, simpler and shorter. Additionally, the lack of statistical significance does not necessarily imply that the differences were unimportant, as there could be a lack of power due to the relatively small numbers of responses of certain categories, e.g., regarding cats. Shelters have an extremely important role in the identification of new infectious agents [16,20,25]. Information on disease occurrence or vaccination collected by questionnaires has limitations as mentioned above. Disease data should be monitored and recorded on a regular basis by shelters [1], and for this purpose, specific shelter management software should be made available in Portuguese, and its use promoted among all shelter types. Ideally, data collection should be harmonized and follow defined diagnostic protocols and case definitions, enabling comparison of the situation in different shelters, detecting outbreaks and improving management, and to contribute overall to animal welfare. 

## 5. Conclusions

The application of prophylactic measures such as vaccinations currently appears to be common in animal shelters that responded to this questionnaire. The majority of shelters vaccinated animals with core vaccines and on an annual basis. This seems to be related to the excessive length of stay of animals in many shelters, an issue requiring urgent intervention, e.g., by increasing shelter staff training and by making better shelter management software available. Infectious diseases still occur, in particular upper respiratory tract infections, parvovirus and dermatophytosis in both dogs and cats. Today, the diagnosis of diseases such as distemper, parvovirus and canine coronavirus should be based on testing and not rely solely on clinical aspects. More accurate information on disease prevalence and incidence could be obtained by objective monitoring systems.

## Figures and Tables

**Figure 1 animals-13-02723-f001:**
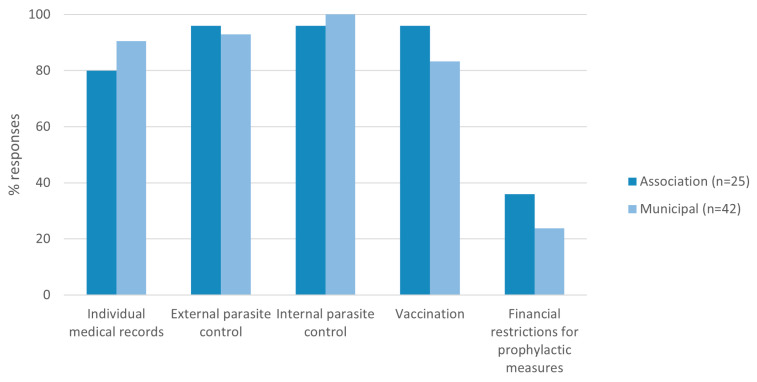
Routine prophylactic practices applied in animal shelters. No statistically significant differences between municipal shelters and associations were determined.

**Figure 2 animals-13-02723-f002:**
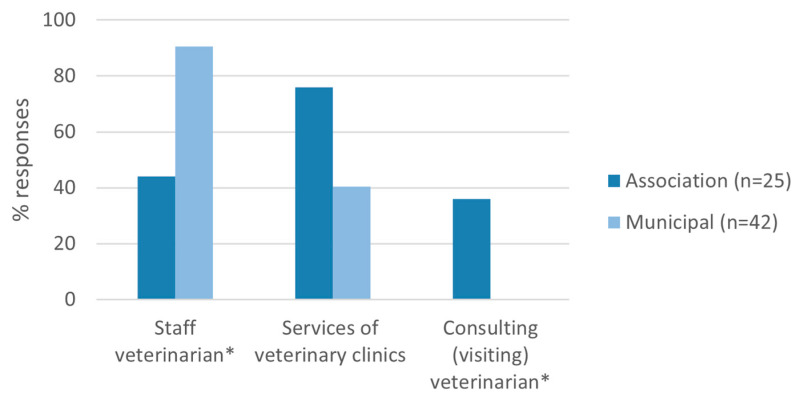
Routine veterinary assistance of animal shelters. More than one option could be selected by the respondents. * Indicates statistically significant difference between municipal shelters and associations (d.f. 1, *p* < 0.05).

**Figure 3 animals-13-02723-f003:**
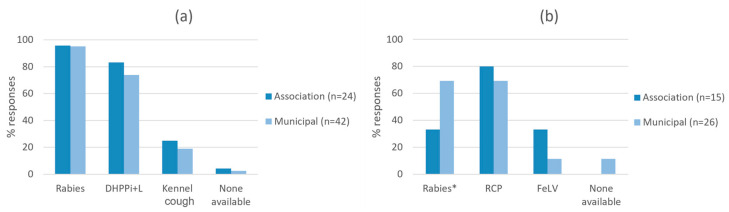
Available vaccines and vaccination cover at animal shelters for (**a**) dogs, and (**b**) cats. * Indicates statistically significant difference between municipal shelters and associations (d.f. 1, *p* = 0.049).

**Figure 4 animals-13-02723-f004:**
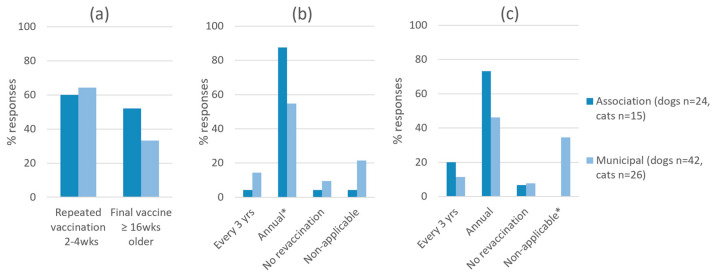
Vaccination schedules at animal shelters: (**a**) Puppy and kitten vaccination, (**b**) Revaccination of adult dogs and (**c**) Revaccination of adult cats. * Indicates statistically significant difference between municipal shelters and associations (d.f. 1, *p* < 0.05).

**Figure 5 animals-13-02723-f005:**
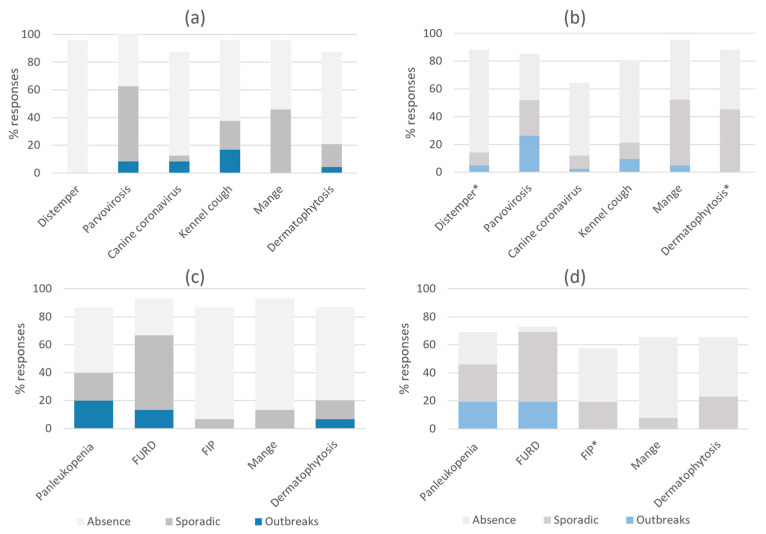
Occurrence of specific infectious diseases within the past 12 months in dogs indicated by (**a**) associations (*n* = 24) and (**b**) municipal shelters (*n* = 42); and in cats as reported by (**c**) associations (*n* = 15) and (**d**) municipal shelters (*n* = 26). The response options “unknown” and “non-applicable” were merged and represent the remainder to 100% of the responses in the graphs. * Indicates statistically significant differences between municipal shelters and associations (d.f. 1, *p* < 0.05); see text for details.

**Figure 6 animals-13-02723-f006:**
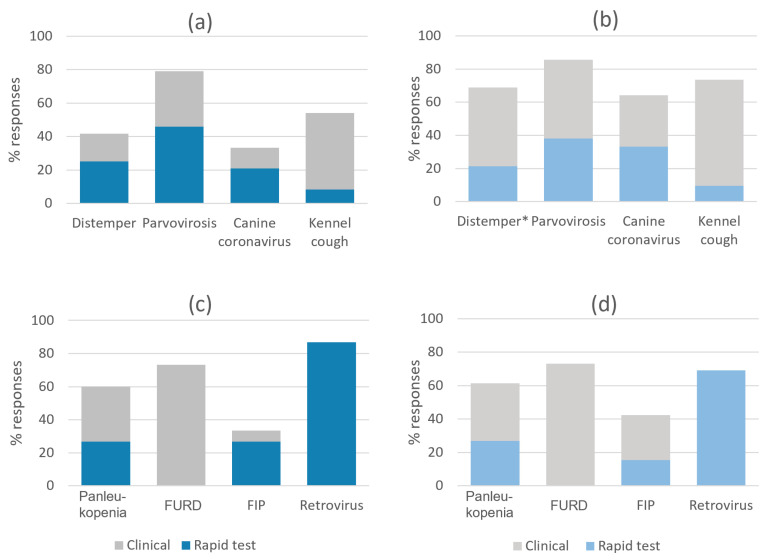
Main diagnostic methods for dogs as indicated by (**a**) associations (*n* = 24) and (**b**) municipal shelters (*n* = 42); and for cats as indicated by (**c**) associations (*n* = 15) and (**d**) municipal shelters (*n* = 26). * Indicates statistically significant difference between municipal shelters and associations (d.f. 1, *p* < 0.05); see text for details.

**Figure 7 animals-13-02723-f007:**
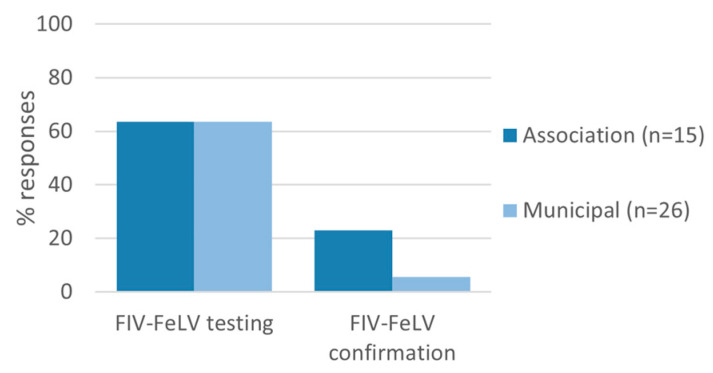
Feline retrovirus diagnosis by rapid tests and positive test confirmation by rapid test or laboratory testing. No statistically significant differences were observed between municipal shelters and associations.

**Table 1 animals-13-02723-t001:** Response rates of animal shelters on the percentage of animals that are in the shelter for more than 1 year.

Animal Species	Response Category	Response Rate	
		Associations %	Municipal %
Dogs	<5%	8 (2/24)	12(5/42)
	5 to 10%	4 (1/24)	10 (4/42)
	>10 to 25%	21 (5/24)	14 (6/42)
	>25 to 50%	13 (3/24)	31 (13/42)
	>50%	54 (13/24)	33 (14/42)
Cats	<5%	13 (2/15)	42 (11/26)
	5 to 10%	7 (1/15)	8 (2/26)
	>10 to 25%	13 (2/15)	12 (3/26)
	>25 to 50%	7 (1/15)	15 (4/26)
	>50% *	60 (9/15)	12 (3/26)
	n.a. **	0 (0/15)	12 (3/26)

* *p* = 0.003; ** n.a. = non-applicable.

## Data Availability

The data presented in this study are available on request from the corresponding authors.

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
