# Peer review of "A Survey on Vaccination and Disease Occurrence in Municipal and Non-Profit Animal Shelters in Portugal"

_animals, 2023, doi:10.3390/ani13172723_

Round 1
Reviewer 1 Report
This is an interesting article which investigates prophylactic measures to prevent disease and subsequent disease occurrence in animal shelters in Portugal. It is of interest to a wider readership and to the field. The study appears well written and includes some interesting points, as well as being well executed with limitations detailed in the discussion
I only have a few minor comments which I have detailed below
Line 27- this is the only real mention of ring worm. Perhaps mention it in the document, or remove it here?
Line 62-64- This is unclear and doesn’t make sense. Please reword. Sorry I cant offer any suggestions to modify it as I don’t know what you are trying to say
Line 155- perhaps at the association shelters rather than at the first?
Line 251- each individual pet, there is no one size … (reword)
Line 268- is this methodology correct for the vaccine used by the shelters? Some are still one year Doi whereas others are 3 so it may be that it is correct for the vaccine in use at the shelter?
Line 345- allowed us to collect some information on. ….. (reword)
Line 351-352- this is unclear and doesn’t make sense. Please reword
Very minor comments detailed above
Author Response
Thank you for your review and clear suggestions. These were very useful to improve the manuscript.
Line 27- this is the only real mention of ring worm. Perhaps mention it in the document, or remove it here?
=> We are using the term “ring worm” only here, in the Simple Summary. This section should be written to the lay audience, whereby we propose to maintain this term here. The more technical term “dermatophytosis” is used throughout the remainder of the text.
Line 62-64- This is unclear and doesn’t make sense. Please reword. Sorry I cant offer anysuggestions to modify it as I don’t know what you are trying to say
=> Thank you, we simplified the text, hoping it is now easier to understand
Line 155- perhaps at the association shelters rather than at the first?
=> Yes, thank you
Line 251- each individual pet, there is no one size … (reword)
=> Thanks, we reworded the sentence.
Line 268- is this methodology correct for the vaccine used by the shelters? Some are still one year Doi whereas others are 3 so it may be that it is correct for the vaccine in use at the shelter?
=> This is true, we corrected this in our text.
Line 345- allowed us to collect some information on. ….. (reword)
=> Thanks, we reworded the sentence.
Line 351-352- this is unclear and doesn’t make sense. Please reword
=> We removed this sentence, as it was only confusing, not adding value to the discussion. Thank you.
Reviewer 2 Report
1. Abstract, line 39 and Results line 198: Regarding "Cat Flu," recommend using the medically correct term for this and any disease/condition at least the first time it's used in the manuscript. For example: ...feline upper respiratory diseases including feline Herpesvirus and Calicivirus, also known collectively as "Cat Flu."
2. Introduction: Was the number of shelters that used standardized quarantine measures assessed?
3. Materials and Methods: Line 99-105: The number and percentage of respondents should probably be listed in the Results section instead of in Methods.
4. Materials and Methods line 108 and Results Line 148/Figure 1: "Internal worming" and "External Worming" would be more accurately described as "Internal Parasite Control" and "External Parasite Control," respectively, since not all internal or external parasites are "worms."
5. Materials and Methods Line 118 and Results Line 199: Regarding "Scabies," recommend that you explain in the Materials and Methods section that you are grouping both Demodex and Sarcoptes as "Scabies," rather than waiting until the Discussion section to do this, so the reader understands this as they begin reading. In some countries, the non-medical term "Scabies" refers ONYl to Sarcoptes where the term "Mange" is used to generalize either Demodex or Sarcoptes, so I went through most of the manuscript errantly wondering why you had not inquired about Demodex.
Materials and Methods Lines 123-124; Results Line 207: Recommend you further clarify and define what you mean by "Clinical diagnosis." While I believe this probably means physical exam and history without diagnostic laboratory tests, some definitions of clinical diagnosis include blood and serologic laboratory tests but are differentiated from histological and other pathological tests. Recommend that you define this more clearly in Materials and Methods.
Discussion Lines 247-249: Does Portugal have restrictions on who can acquire and administered animal vaccinations? For example, in some countries canine distemper-parvo and feline RCP can be purchased and administered by anyone without a veterinary prescription, but rabies can only be administered under the direct supervision of a veterinarian or by a certified rabies vaccination administrator, which is more common in Municipal shelters. If this is also true in Portugal, then maybe the higher number of cats vaccinated against rabies in municipal shelters corresponds with the higher number of veterinarian respondents in Municipal shelters. It would be interesting to know if there was a difference between Municipal and Association shelters in whether they had a staff veterinarian, consulting (visiting) veterinarian, or relied solely on the services of veterinary clinics within the community, as this might impact the access to veterinary care and preventive measures in place at each shelter. is it at all possible to go back and obtain that information from the respondents?
Discussion Line 306: Authors have used the term "panleukopenia" up until line 306, then refer to it as feline parvovirus. Recommend using the same term throughout the manuscript, with possible mention of the parvovirus term, for example: ...feline panleukopenia, also known as feline parvovirus.
Use of English language is excellent, perhaps with a few minor typographical errors or differences in non-medical names for diseases (Scabies meaning both Demodex and Sarcoptic mange, vs. Scabies meaning only Sarcoptes in English)
Author Response
Thank you for your review and clear suggestions. These were very useful to improve the manuscript.
- Abstract, line 39 and Results line 198: Regarding "Cat Flu," recommend using the medically correct term for this and any disease/condition at least the first time it's used in the manuscript. For example: ...feline upper respiratory diseases including feline Herpesvirus and Calicivirus,also known collectively as "Cat Flu."
=> We corrected this, and also completed the definition regarding canine infectious tracheobronchitis (kennel cough).
- Introduction: Was the number of shelters that used standardized quarantine measures assessed?
=> Unfortunately not. This is would have been interesting indeed.
- Materials and Methods: Line 99-105: The number and percentage of respondents should probably be listed in the Results section instead of in Methods.
=> We moved this paragraph to the Results section.
- and"External Worming" would be more accurately described as "Internal Parasite Control" and"External Parasite Control," respectively, since not all internal or external parasites are "worms."
=> Yes, thank you. We also updated Figure 1 accordingly. We left the term “worming” in the simple summary which is destined to a general audience.
- Materials and Methods Line 118 and Results Line 199: Regarding "Scabies," recommend that you explain in the Materials and Methods section that you are grouping both Demodex and Sarcoptes as "Scabies," rather than waiting until the Discussion section to do this, so the reader understands this as they begin reading. In some countries, the non-medical term "Scabies"refers ONYl to Sarcoptes where the term "Mange" is used to generalize either Demodex orSarcoptes, so I went through most of the manuscript errantly wondering why you had not inquired about Demodex.
=> Thank you for this explanation. Accordingly, we decided to use the term “mange” instead of “scabies”, and explained this in Material and Methods, and also mentioned it in the Discussion, in the limitations of our study.
We also updated Figures 4 and 5 accordingly (use “mange”; “panleukopenia” instead of “FPV”)
- Materials and Methods Lines 123-124; Results Line 207: Recommend you further clarify and define what you mean by "Clinical diagnosis." While I believe this probably means physical exam and history without diagnostic laboratory tests, some definitions of clinical diagnosis include blood and serologic laboratory tests but are differentiated from histological and other pathological tests. Recommend that you define this more clearly in Materials and Methods.
=> Yes, we now defined the term “clinical diagnosis” in Material and Methods.
- Discussion Lines 247-249: Does Portugal have restrictions on who can acquire and administered animal vaccinations? For example, in some countries canine distemper-parvo and feline RCP can be purchased and administered by anyone without a veterinary prescription, but rabies can only be administered under the direct supervision of a veterinarian or by a certified rabies vaccination administrator, which is more common in Municipal shelters. If this is also true in Portugal, then maybe the higher number of cats vaccinated against rabies in municipal shelters corresponds with the higher number of veterinarian respondents in Municipal shelters.
=> Vaccination of animals is indeed a medical act and requires supervision by a veterinarian if carried out by others. Thus, we think the higher number of cats vaccinated against rabies in municipal shelters is unrelated to the higher number of veterinarians responding for Municipal shelters.
- It would be interesting to know if there was a difference between Municipal and Association shelters in whether they had a staff veterinarian, consulting (visiting) veterinarian, or relied solely on the services of veterinary clinics within the community, as this might impact the access to veterinary care and preventive measures in place at each shelter. is it at all possible to go back and obtain that information from the respondents?
=> Yes, we also had this question in the questionnaire, so we included it now in the manuscript (Material& Methods, Results Figure 2)
- Discussion Line 306: Authors have used the term "panleukopenia" up until line 306, then refer to it as feline parvovirus. Recommend using the same term throughout the manuscript, with possible mention of the parvovirus term, for example: ...feline panleukopenia, also known as feline parvovirus.
=> Yes, thank you, was updated accordingly.
Reviewer 3 Report
Title: please add the country for this work. Also, that this includes both government run (municipal) and non-profit organizations.
Please round all percentages to whole numbers and include actual p-values. And round p-values to 1 or 2 significant digits (so a p-value of 0.0486 would be either 0.05 or 0.049 and then be consistent throughout the manuscript.
Abstract: is cat flu the feline upper respiratory infection (URI) viruses infection?
Line 30: the research isn’t about overcrowding per se. And shelters can have poor preventive protocols and not be overcrowded. Please edit both of these sentences to better reflect that.
Introduction: I think that this manuscript is about learning more about Portugues animal shelters relative to the ordinance as well as relative to recommendations for basic preventive care? Please clarify in introduction. And include in the introduction what the standards are for shelters in Portugal and more generally for this type of preventive care. Parts of the manuscript read as if the purpose is to state that shelters in Portugal are doing a great job. Given the limitations of this study and the actual findings I don’t think that is true. However, the shelters may be doing as well as anywhere else—and there isn’t a lot of data about how shelters actually do conform to guidelines (or laws). Please be clear about the purposes of this manuscript and check that the tone and content focus on those purposes.
Why would shelters be doing annual vaccinations? Unless they are sanctuaries or have extremely long hold times, this isn’t likely to be critically important. This should be explained in the introduction also.
What are the standards for shelters in Portugal? Is there something that is published? This study feels like the authors are trying to show that shelters are doing a good job but without a set of standards or expectations, that is difficult to do.
Based on the analysis, the comparison between municipal shelters and associations was the critical element. But no information about why this is important in Portugal is here. And not enough discussion and conclusion is present about these differences and what should be done about them.
Line 53-4: is there more detail about what parasite control and vaccination needs to be done? Only on leaving or while there? Some additional translation of the ordinance would be helpful for the context.
Line 56: I’m not sure what non-profits organized as associations means. And please add the definition of municipal shelters as well. Are both of these types of organizations found in a dedicated physical building or are some also based on fostering the animals in private homes (or some other system)? Question A3 looks like it separates non-profits into those that do adoptions and sanctuaries which do not. That is potentially an important difference relative to preventive care and disease management.
Sentence ending on line 74 please reference.
Line 76: and to vaccinate at intake to the shelter. Please add to text.
Line 79: is this meant to refer to Portugal or all animal shelters? Please clarify in the text.
Paragraph starting on line 80: please clarify in the text what countries there are from…and only include if the animal sheltering system in Portugal is likely to be similar. Otherwise consider describing how the systems are likely different here. And in the discussion describe why these differences might be important and make the existing research not applicable to Portugal.
Section 2.1: any analysis or data that came from the questionnaire or was about the questionnaire goes into the results. And lack of statistical significance doesn’t mean that the differences are unimportant. There could be a lack of power. Please include that in the discussion and consider some post-hoc power analysis where there are clinically important differences which turned out to be not statistically significant. I don’t think that the response rate differences are clinically significant for example. Were there instructions for the questionnaire about who to have complete it or why it was being done? That isn’t included in the supplement and any materials shared with the questionnaire should be added to the text.
Line 108: I believe the better translation for “individual medical sheets” is “individual medical records”.
Section 2.2: this is where the tests which were run should be listed. And I think that there is a stats test missing as the median number of animals would not be compared via chi-square. Please check and edit accordingly in the text.
Line 117: based on information later in the manuscript, scabies would actually be translated better as “mange” which is non-specific to mite species (or even to knowing for sure it is a mite)
Section 3.1 here is where data about how many Associations were sanctuaries should be and if there were any “other” responses. Please include data for all responses to each question.
And please include the number in the numerator and denominator as well as the percentages. Otherwise, to see how many associations provided annual vaccinations for cats requires taking 73% and multiplying by the 26 shelters. Instead in the text write 73% (19/26) and similarly throughout the manuscript.
Figure 2a so Felv differences between the shelter types was not tested or not run? And was one chi-square test run for all dog vaccines or one test for each type of vaccine? Please clarify in methods and results what was compared to what.
Line 163 and following: I think this is for both puppies and kittens together?
Line 172-3: is this because these associations were sanctuaries?
Figure 4, not all diseases reach 100%, based on the text they should? Please correct. And if the primary comparison is by type of shelter, these graphs should also be arranged like figure 3; instead of comparing all dog diseases in one graph, there should be bars for each type of shelter next to each other for a given dog disease. So essentially a and b would be overlayed. Same for Figure 5.
Line 238-9: were those shelters in need municipal or non-profit? That is important as funding sources are different and require different mechanisms to increase funding.
Line 248-9: are there possibly local mandates for rabies vaccination for cats? Please edit the text and include any information on this.
Line 257: again, this is for puppies and kittens isn’t it (looking at the questionnaire).
Paragraph starting on line 267: I don’t understand why any shelter that isn’t a sanctuary would anticipate annual vaccinations unless it is typical for shelters to hold animals for more than one year. I’m assuming from other statements in the manuscript that the length of stay is unknown. That should be included in this paragraph so that the context for the results about annual vaccination is clear.
Line 286-7: was there a difference in testing between shelter types? I can’t tell from this sentence if that was true or not.
Lines 333-4: I’m not sure that reference 9 is accurate anymore given the new understanding of FeLV.
Line 350-1: I’m not sure what this sentence means: The questionnaire structure presented too many 350 questions that made analyses difficult.
Line 352-3: Actually, the next questionnaire should be designed by people with a lot of experience in this area, then evaluated by pilot testing and cognitive interviewing. The survey was rather short already, so I don’t think that was the issue. It is also not clear how many reminders were sent, to whom, and by what route, all of which are important for response rates. And even so, the response rate is pretty good!
Additional limitations: limitation about the vaccination frequency: question is leading “check if do puppy kitten vaccines at 2-4 weeks”, “check if do final at 16 weeks”. The only option is “other”. And outbreak is defined as >2 cases at the same time period. In this survey an "outbreak" is considered to be 2 or more animals affected in the same period. So how different from >2? I see another answer option but I’m not sure of the translation. Or then how the different categories were clearly defined to be distinct.
356-61: How does this fit with the focus of this manuscript? Is this the next study? Or more for what kinds of education is needed for animal sheltering in Portugal? Based on what?
Conclusions: given that vaccination on intake is standard protocol (it is a must in the ASV guidelines pg. 32) and some shelters can’t due to financial constraints I’d say there is still quite a bit of room for this to improve. And even an individual shelter software system (are there some good ones in Portugal?) is critically important before anything national can be done.
I have made a few suggestions for translation of some terminology. There are a number of sentences that aren't clear enough to understand (also identified).
It is a little difficult to suggest content edits because of some awkward language and hard to determine how much is language-based or due to limited familiarity with writing manuscripts for publication.
And some sections are better written both in clarity of English and in organization and sentence structure.
Author Response
Thank you for your very thorough review and suggestions. Some of your comments were quite challenging and prompted us to re-think several aspects of our work. Your feedback was very useful to improve the manuscript.
1) Title: please add the country for this work. Also, that this includes both government run (municipal) and non-profit organizations.
=> The title was completed accordingly. The reason for not doing so initially was to keep the title short.
2) Please round all percentages to whole numbers and include actual p-values. And round p-values to 1 or 2 significant digits (so a p-value of 0.0486 would be either 0.05 or 0.049 and then be consistent throughout the manuscript.
=> Yes, thank you. This was changed.
3) Abstract: is cat flu the feline upper respiratory infection (URI) viruses infection?
=> This was changed in the abstract, and also in Material & Methods.
4) Line 30: the research isn’t about overcrowding per se. And shelters can have poor preventive protocols and not be overcrowded. Please edit both of these sentences to better reflect that.
=> Yes, thank you. This was changed.
5a) Introduction: I think that this manuscript is about learning more about Portugues animal shelters relative to the ordinance as well as relative to recommendations for basic preventive care? Please clarify in introduction. And include in the introduction what the standards are for shelters in Portugal and more generally for this type of preventive care.
=> We have update the introduction accordingly.
- b) Parts of the manuscript read as if the purpose is to state that shelters in Portugal are doing a great job. Given the limitations of this study and the actual findings I don’t think that is true. However, the shelters may be doing as well as anywhere else—and there isn’t a lot of data about how shelters actually do conform to guidelines (or laws). Please be clear about the purposes of this manuscript and check that the tone and content focus on those purposes.
=> We were surprised by your interpretation, and we agree that shelters in Portugal have to improve substantially (see also below). Some aspects may have considerably improved in the past decades, although this is based on personal experience of shelter vets, not on objective data. Others aspects require improvement. Too little is known, really, so with this questionnaire we wished to start somehow, to get a baseline of the situation. We modified parts of the manuscript accordingly.
6) Why would shelters be doing annual vaccinations? Unless they are sanctuaries or have extremely long hold times, this isn’t likely to be critically important. This should be explained in the introduction also.
=> They do have extremely long hold times and we think this is a major problem in Portugal. We published some data as poster, and were unsure whether to include this information here, but as you raised this issue, we did.
7) What are the standards for shelters in Portugal? Is there something that is published? This study feels like the authors are trying to show that shelters are doing a good job but without a set of standards or expectations, that is difficult to do.
=> See above. There are no general shelter standards, just legal requirements for animals being adopted from Municipal shelters.
8) Based on the analysis, the comparison between municipal shelters and associations was the critical element. But no information about why this is important in Portugal is here. And not enough discussion and conclusion is present about these differences and what should be done about them.
=> This is due to the collaboration between both. This is now explained in the introduction.
9) Line 53-4: is there more detail about what parasite control and vaccination needs to be done? Only on leaving or while there? Some additional translation of the ordinance would be helpful for the context.
=> This was completed in the introduction
10) Line 56: I’m not sure what non-profits organized as associations means. And please add the definition of municipal shelters as well. Are both of these types of organizations found in a dedicated physical building or are some also based on fostering the animals in private homes (or some other system)? Question A3 looks like it separates non-profits into those that do adoptions and sanctuaries which do not. That is potentially an important difference relative to preventive care and disease management.
=> The definitions and explanations of the different shelter types was now included in Introduction, Methods and Results. The sanctuaries started to respond but, as they did not complete the questionnaire, they were excluded from analysis
11) Sentence ending on line 74 please reference.
=> References were added.
12) Line 76: and to vaccinate at intake to the shelter. Please add to text.
=> This was added, thank you.
13) Line 79: is this meant to refer to Portugal or all animal shelters? Please clarify in the text.
=> Yes, the sentence was completed accordingly.
14) Paragraph starting on line 80: please clarify in the text what countries there are from…and only include if the animal sheltering system in Portugal is likely to be similar. Otherwise consider describing how the systems are likely different here. And in the discussion describe why these differences might be important and make the existing research not applicable to Portugal.
=> Most published studies were carried out in the US. We do not know if the situation is likely to be similar in Portugal and elsewhere in Europe, because there are only few publications from Europe, and less even from Portugal.
15a) Section 2.1: any analysis or data that came from the questionnaire or was about the questionnaire goes into the results.
=> This was changed accordingly
- b) And lack of statistical significance doesn’t mean that the differences are unimportant. There could be a lack of power. Please include that in the discussion and consider some post-hoc power analysis where there are clinically important differences which turned out to be not statistically significant. I don’t think that the response rate differences are clinically significant for example.
=> We agree and included the lack of power in the discussion, as limitation of this study.
We did not discuss the clinical relevance of different response rates as our study did not evaluate this.
- c) Were there instructions for the questionnaire about who to have complete it or why it was being done? That isn’t included in the supplement and any materials shared with the questionnaire should be added to the text.
=> The text in Material and Methods was changed accordingly
=> Instructions on who should fill in the questionnaire were given in the cover letter. One of these letters was now added to the questionnaire and made available as Supplementary Information.
16) Line 108: I believe the better translation for “individual medical sheets” is “individual medical records”.
=> Thank you, this was corrected accordingly in the text and in the figure legend (Fig 19).
17) Section 2.2: this is where the tests which were run should be listed. And I think that there is a stats test missing as the median number of animals would not be compared via chi-square. Please check and edit accordingly in the text.
=> Thank you, this was completed accordingly
18) Line 117: based on information later in the manuscript, scabies would actually be translated better as “mange” which is non-specific to mite species (or even to knowing for sure it is a mite)
=> Thank you, this was corrected accordingly
19) Section 3.1 here is where data about how many Associations were sanctuaries should be and if there were any “other” responses. Please include data for all responses to each question.
=> As mentioned above, the 3 sanctuaries were excluded from analysis as they did not complete the questionnaire.
20) And please include the number in the numerator and denominator as well as the percentages. Otherwise, to see how many associations provided annual vaccinations for cats requires taking 73% and multiplying by the 26 shelters. Instead in the text write 73% (19/26) and similarly throughout the manuscript.
=> Numerator/denominator/percentages were completed.
21) Figure 2a so Felv differences between the shelter types was not tested or not run? And was one chi-square test run for all dog vaccines or one test for each type of vaccine? Please clarify in methods and results what was compared to what.
=> One test was run for each vaccine type, also for FeLV (p= 0.12). We did clarify this in methods and results.
22) Line 163 and following: I think this is for both puppies and kittens together?
=> Correct, the text in materials and results was amended accordingly.
23) Line 172-3: is this because these associations were sanctuaries?
=> None were sanctuaries, see above.
24a) Figure 4, not all diseases reach 100%, based on the text they should? Please correct.
=> This gap represents “unknown/non applicable”, initially explained in lines 179-183 of the Results, we have now included this information in the Materials and also in the figure legend.
- b) And if the primary comparison is by type of shelter, these graphs should also be arranged like figure 3; instead of comparing all dog diseases in one graph, there should be bars for each type of shelter next to each other for a given dog disease. So essentially a and b would be overlayed. Same for Figure 5.
=> Yes, we agree and did these figures, but then had to double the legends, and preferred to maintain the current graphs.
25) Line 238-9: were those shelters in need municipal or non-profit? That is important as funding sources are different and require different mechanisms to increase funding.
=> No statistically significant difference was found between shelter types (See Results). We reworded the sentence to be clearer.
26) Line 248-9: are there possibly local mandates for rabies vaccination for cats? Please edit the text and include any information on this.
=> There are no local mandates for rabies vaccination, as Portugal is free from terrestrial rabies since 1961. This was now included in the manuscript.
27) Line 257: again, this is for puppies and kittens isn’t it (looking at the questionnaire).
=> Yes, of course, this was corrected.
28) Paragraph starting on line 267: I don’t understand why any shelter that isn’t a sanctuary would anticipate annual vaccinations unless it is typical for shelters to hold animals for more than one year. I’m assuming from other statements in the manuscript that the length of stay is unknown. That should be included in this paragraph so that the context for the results about annual vaccination is clear.
=> See above, we now included some proxy on length of stay
29) Line 286-7: was there a difference in testing between shelter types? I can’t tell from this sentence if that was true or not.
=> True, this sentence was removed.
30) Lines 333-4: I’m not sure that reference 9 is accurate anymore given the new understanding of FeLV.
=> This reference was updated to Westmann et al. 2019.
31) Line 350-1: I’m not sure what this sentence means: The questionnaire structure presented too many questions that made analyses difficult.
=> The questionnaire contained 11 sections and some results were already published. This information was now included in the Methods.
32) Line 352-3: Actually, the next questionnaire should be designed by people with a lot of experience in this area, then evaluated by pilot testing and cognitive interviewing. The survey was rather short already, so I don’t think that was the issue. It is also not clear how many reminders were sent, to whom, and by what route, all of which are important for response rates. And even so, the response rate is pretty good!
=> The questionnaire was developed by academic staff and two municipal shelter veterinarians. One vet is from one of the largest Municipal Shelter in the country, and both have more than 20 years of work experience, but lack formal (academic/scientific) training on the subject, such as most of their shelter colleagues.
=> Shelter Medicine in Portugal is in its infancy and only in the past few years gained attention by the veterinary profession. In response, Universities are now starting to offer Continuing Education courses => We have included this in the discussion – thank you.
=> The questionnaire was tested previously by 5 shelter veterinarians and their feedback incorporated only minor suggestions, so we assumed the quality of the questionnaire would be ok. In hindsight, the questionnaire should have been much simpler.
=> Also, reminders were sent by email, and we have now included this information in the Materials.
33a) Additional limitations: limitation about the vaccination frequency: question is leading “check if do puppy kitten vaccines at 2-4 weeks”, “check if do final at 16 weeks”. The only option is “other”.
=> This is probably a misunderstanding: Puppy vaccination is “every 2-4 weeks, final vaccine at 16 weeks or older” (in Methods). We have changed this to “repeated every 2-4 weeks, final vaccination at 16 weeks or older”, hope it is clearer now
- b) And outbreak is defined as >2 cases at the same time period. In this survey an "outbreak" is considered to be 2 or more animals affected in the same period. So how different from >2? I see another answer option but I’m not sure of the translation. Or then how the different categories were clearly defined to be distinct.
=> We have completed the sentence “Cases were defined … at different time points….”
34) 356-61: How does this fit with the focus of this manuscript? Is this the next study? Or more for what kinds of education is needed for animal sheltering in Portugal? Based on what?
=> We have partly rewritten the discussion and also included comments on Length of stay and the current development of shelter continuing professional development
35) Conclusions: given that vaccination on intake is standard protocol (it is a must in the ASV guidelines pg. 32) and some shelters can’t due to financial constraints I’d say there is still quite a bit of room for this to improve. And even an individual shelter software system (are there some good ones in Portugal?) is critically important before anything national can be done.
=> Yes, we agree that there is consirable room for improvement, and we have included this in the manuscript.
=> There are, to the best of our knowledge, only few shelter software available in Portugal, and these are quite basic. Some shelters use software designed for Veterinary Clinics, which are also of limited use to manage shelters.
Round 2
Reviewer 2 Report
Thank you for addressing my questions and concerns. This reads more clearly for me now and I believe will have the same effect for readers.
Author Response
Thank you for addressing my questions and concerns. This reads more clearly for me now and I believe will have the same effect for readers.
=> Thank you.
Reviewer 3 Report
The authors have done a very good job of clarifying the focus, analysis, and results of the manuscript.
Line 248: please add a comment that the survey and results were during Covid so that context is clear.
Lines 406 & 409 please round these percentages also.
Lines 453-6: Thank you for including these data. I understand that this is the definition of excessive but think that adding the number and percent for all categories of responses here would also be helpful.
Figure 3: please update the legend with the actual p-value
Figure 4a: is puppies and kittens, please edit.
Line 690-2: Length of stay is also likely related to adoption or other live outcome limitations. If shelter resources are consumed by caring for the existing animals, increasing adoptions and outreach to do that are likely under-emphasized. In the US we’ve also found that controlling intake rate and timing of intake can help free up resources to better care for animals and increase live outcomes. It doesn’t sound like managing intake is an option for Portugal. And if euthanasia for space rather than illness, injury or danger to others from behavior isn’t an option, the challenge is even greater. Italy has this same issue. Please consider adding some of these ideas to this section. All of the flows in and out of shelters are related and a decrease or increase in one leads to changes in the others.
Lines 732-3: and treating dermatophytosis to clear the infection is extremely lengthy adding to cost and time in shelter particularly of potentially more adoptable kittens and puppies.
Reply #32 and #33a: Please don’t think that I am being overly critical. There is both an art and a science to writing questionnaires which require special training and experience, in any language. Someone with research experience in questionnaire design could have provided useful insights into designing the questions. Please add in 2.1 that 5 other shelter veterinarians who did not design the questionnaire also reviewed the survey and that their minor comments were incorporated. My observation about the vaccination frequency being a “leading” question is that the options were primarily about the “correct” answer. So it was easier for them to say “yes, this is what we do”. To really understand the vaccination patterns the question would have had to provide a list of other options or some open-ended text. For example, “please indicate how often you vaccinate puppies and kittens after they arrive at your shelter: once, every 2 weeks, every 3-4 weeks, every 5-8 weeks, other. At what age do you stop the initial series of puppy/kitten vaccines? 1 month of age, 2 months of age, 3 months of age, 4 months of age, 5 months of age, other.” That way the response would be less likely to be the one considered to be the “correct” one. I understand about having to estimate ages but puppies and kittens are relatively easy to age up to 6 months or so.
The information is understandable, but there are still some English language errors. Up to the editor to determine if they need to be addressed.
Author Response
=> Thank you, again, for your thorough review and clear suggestions. These were very useful to improve the manuscript.
The authors have done a very good job of clarifying the focus, analysis, and results of the manuscript.
Line 248: please add a comment that the survey and results were during Covid so that context is clear.
=> Yes, this was added in the Materials Section “2.1 Questionnaire”
Lines 406 & 409 please round these percentages also.
=> The percentages were rounded.
Lines 453-6: Thank you for including these data. I understand that this is the definition of excessive but think that adding the number and percent for all categories of responses here would also be helpful.
=> Thank you for this suggestion. We added a table with the number and percentage for all categories and introduced a sentence in the paragraph underneath Figure 2.
Figure 3: please update the legend with the actual p-value
=> Thank you, we have updated the legend accordingly.
Figure 4a: is puppies and kittens, please edit.
=> Thank you, we edited the legend accordingly.
Line 690-2: Length of stay is also likely related to adoption or other live outcome limitations. If shelter resources are consumed by caring for the existing animals, increasing adoptions and outreach to do that are likely under-emphasized. In the US we’ve also found that controlling intake rate and timing of intake can help free up resources to better care for animals and increase live outcomes. It doesn’t sound like managing intake is an option for Portugal. And if euthanasia for space rather than illness, injury or danger to others from behavior isn’t an option, the challenge is even greater. Italy has this same issue. Please consider adding some of these ideas to this section. All of the flows in and out of shelters are related and a decrease or increase in one leads to changes in the others.
=> Thank you, we added these ideas into the discussion.
Lines 732-3: and treating dermatophytosis to clear the infection is extremely lengthy adding to cost and time in shelter particularly of potentially more adoptable kittens and puppies.
=> Thank you, this is so true. We updated the sentence.
Reply #32 and #33a: Please don’t think that I am being overly critical. There is both an art and a science to writing questionnaires which require special training and experience, in any language. Someone with research experience in questionnaire design could have provided useful insights into designing the questions. Please add in 2.1 that 5 other shelter veterinarians who did not design the questionnaire also reviewed the survey and that their minor comments were incorporated. My observation about the vaccination frequency being a “leading” question is that the options were primarily about the “correct” answer. So it was easier for them to say “yes, this is what we do”. To really understand the vaccination patterns the question would have had to provide a list of other options or some open-ended text. For example, “please indicate how often you vaccinate puppies and kittens after they arrive at your shelter: once, every 2 weeks, every 3-4 weeks, every 5-8 weeks, other. At what age do you stop the initial series of puppy/kitten vaccines? 1 month of age, 2 months of age, 3 months of age, 4 months of age, 5 months of age, other.” That way the response would be less likely to be the one considered to be the “correct” one. I understand about having to estimate ages but puppies and kittens are relatively easy to age up to 6 months or so.
=> Yes, we agree, that it would have been extremely useful to include someone with experience in writing questionnaires. We appreciate your suggestions on the formulation of the questions. Definitely, this is an area we will seek further training to improve our skills.
=> We completed section 2.1. with information on the five shelter veterinarians that reviewed the questionnaire.